# Risk of Avascular Necrosis with The Modified Dunn Procedure in SCFE Patients: A Meta-Analysis

**DOI:** 10.3390/children9111680

**Published:** 2022-10-31

**Authors:** Julio J. Jauregui, Nichole M. Shaw, Tristan B. Weir, Sherwin A. Barvarz, Philip K. McClure

**Affiliations:** 1Department of Orthopaedics, University of Maryland Medical Center, Baltimore, MD 21201, USA; 2Department of Orthopaedic Surgery and Rehabilitation, SUNY Downstate Medical Center, Brooklyn, NY 11203, USA; 3Rubin Institute for Advanced Orthopaedics, Baltimore, MD 21215, USA

**Keywords:** Slipped Capital Femoral Epiphysis (SCFE), pediatric orthopaedics, pediatric hip

## Abstract

In situ stabilization is a widely accepted treatment for slipped capital femoral epiphysis (SCFE) despite risks of avascular necrosis (AVN) and femoroacetabular impingement (FAI). The modified Dunn procedure with surgical hip dislocation attempts to maintain epiphyseal perfusion and allows anatomic epiphyseal repositioning, theoretically reducing AVN and FAI risks. We systematically evaluated the literature, elucidating overall and stability-stratified rates of AVN following the modified Dunn procedure, and revision rates in non-AVN patients. Using Ovid and MEDLINE (PubMed), studies involving the modified Dunn procedure were evaluated for age, stability, preoperative slip (Southwick) angle, ROM at follow-up, outcome metrics, and revisions. Utilizing a random effect model of proportions, we determined overall and stability-stratified AVN rates, and revision rates in patients without AVN.673 patients (688 SCFEs) who underwent modified Dunn procedure were included. Overall AVN rate was 14.3% with a 95% Confidence Interval (CI) of 9.3 to 20.2%. AVN rate in stable slips was 10.9% (95% CI: 6.0 to 17.1%) and 19.9% (95% CI: 12.8% to 28.1%) in unstable slips. Revision rate in non-AVN patients was 13.3% (95% CI: 8.3% to 19.2%). Fixation failures occurred following K-wire or small-caliber (<6.5 mm) screw fixation. Overall mean Harris Hip Score (HHS) was excellent (>90 points). Mean HHS was 98.9 points (range of means: 86 to 99 points) in stable cases, and 90.5 points (range of means: 73 to 98 points) in unstable cases. Patients undergoing modified Dunn procedure had excellent clinical outcomes and low incidences of AVN. Further studies are needed to determine if modified Dunn osteotomy with surgical hip dislocation is a viable alternative to in situ pinning for treatment of severe SCFE.

## 1. Introduction

The current treatment for slipped capital femoral epiphysis (SCFE) is surgical stabilization of the physis with in situ pinning [1,2,3]. With this treatment, the risk for developing avascular necrosis (AVN) is about 0–1% in patients who have a stable slip as defined by Loder et al. [4]. Additionally, several studies have reported overall good clinical outcomes when managed with this treatment modality [5]. However, as studies with longer follow-up have emerged, numerous authors have described sub-optimal patient-reported clinical outcomes with a potential accelerated progression to osteoarthritis [6,7,8]. Complicating this finding, studies have shown associations between in situ percutaneous pinning and poorer health outcomes compared to the general population, which may further compound the negative outcomes in SCFE treatment [9]. As the anatomy of the proximal femur is often disrupted, femoroacetabular impingement (FAI) can also develop, which can eventually lead to pain, severe functional limitations, and osteoarthritis [7,8,10]. The deformity and debilitating pain may progress to such a degree that it will eventually necessitate a total hip arthroplasty (THA) [3,11]. Therefore, while in situ fixation remains the most commonly used method to treat SCFE patients, the optimal surgical treatment for SCFE remains undetermined [1]. The long term outcome for hips with significant deformity is likely very poor, with very high conversion rates to THA at 40–50 years follow up [12].

In order to correct the residual deformity in SCFE, and thus prevent FAI, current surgical treatment modalities increasingly focus on both stabilizing the “slipped” epiphysis and restoring anatomic alignment of the epiphysis, which remains in the acetabulum while the metaphysis displaces anteriorly and externally rotates [13]. Delayed osteotomies away from the apex of deformity that allow for correction of the anatomic deformity caused by severe SCFE have been described, but can pose additional problems, such as femoral head AVN, and do not fully restore anatomic alignment, as a double level compensatory deformity is created [14,15,16,17,18,19]. Leunig et al. [13] described a technique of epiphyseal realignment by utilizing a modified Dunn osteotomy with a surgical hip dislocation. Many advantages have been described with this procedure, including (1) the ability to potentially maintain epiphyseal perfusion intraoperatively to minimize the risk of AVN, and (2) anatomically reposition the epiphysis on the femoral neck to theoretically minimize the risk of FAI [13,14,15,20,21]. This procedure is technically demanding and may increase the odds of certain complications such as AVN of the femoral head [13].

Although potentially a successful procedure, there is a paucity of studies describing the outcomes and definitively establishing the rate of complications for the modified Dunn osteotomy with a surgical hip dislocation for the treatment of SCFE. Additionally, the literature describes a wide range for the incidence of AVN [13,22]. Therefore, the purpose of this study was to assess the overall rate of AVN with the modified Dunn osteotomy and surgical hip dislocation by systematically reviewing the existing literature. By performing a systematic evaluation of all current literature, we attempted to (1) determine the overall and subset-specific rates of AVN in patients who underwent a modified Dunn procedure, (2) determine the percentage of patients without AVN who required revision surgery, and (3) determine the mean Harris Hip Scores of this patient population after undergoing the modified Dunn procedure.

## 2. Materials and Methods

Following the Preferred Reporting Items for Systematic Reviews and Meta-Analyses (PRISMA) guidelines, we performed a literature search using the Ovid and MEDLINE (PubMed) search engines to determine all studies in which a Modified Dunn osteotomy was used to manage a patient with SCFE [23]. Using the search strings: “slipped[ti] AND capital[ti] AND femoral[ti] AND epiphysis[ti]” and “SCFE[ti]” we found a total of 1164 articles published through July 2021 (Figure 1). We then excluded articles published in languages other than English, obtaining a total of 1096 studies. Additional exclusion criteria included expert opinions, studies describing different osteotomies for the sequela of SCFE, those who had a different primary procedure, studies not performed in humans, and studies with a mean follow-up of less than 12 months.

Following this inclusion/exclusion process, 274 articles remained which were read in full and were all cross-referenced. Cross-referencing did not yield additional studies. Review papers and studies involving osteotomies technically different from the original Modified Dunn osteotomy with surgical dislocation and capital realignment were then excluded, yielding a total of 29 studies. We then excluded a single case report and one other study that only had one patient in the modified Dunn group. Finally, a follow-up study on a previously published dataset was included in the analysis, while the earlier study was excluded. The final 26 studies that evaluated the modified Dunn procedure for the treatment of SCFE were assessed in detail and were part of our final cohort.

In all the evaluated studies, we first determined the number of cases and patients (Appendix A). We then determined the patients who had a stable or unstable SCFE, utilizing the classification system described by Loder et al. [4]. Demographic characteristics such as age and follow-up were also recorded. Range of motion (ROM) in degrees at latest follow-up was evaluated. Specifically, we recorded flexion, internal rotation in flexion (IR), and external rotation in flexion (ER) from all studies. The mean displacement (slip) in degrees preoperatively, as defined by the Southwick angle, was documented for studies in which this information was provided [17].

In terms of specific endpoints, we determined the total number of hips that developed AVN and stratified the number of hips that developed AVN according to their stability as defined by Loder et al. Then, after excluding patients who developed AVN, we determined and quantified the number of revisions required. Finally, we determined the number of AVN patients who underwent total hip arthroplasty within the available follow-up period.

Regarding clinical outcomes, we evaluated all studies in detail to determine the outcome metric utilized. Studies reported the University of California Los Angeles activity score (UCLA), Western Ontario and McMaster Universities Arthritis Index (WOMAC) score, Merle d’Aubigné score, and Harris Hip Score (HHS) (Appendix A). For studies in which a modified HHS was used (range of 0–91 points instead of 0–100 points), we adjusted the reported values to a 100-point scale to standardize the findings. Although we attempted to evaluate each outcome metric score, only the HHS were commonly reported, hence we based our outcomes on this metric. A score greater than 90 out of 100 points was classified as excellent.

All data were compiled onto an electronic spreadsheet (Microsoft Excel, Microsoft Office, Redmond, WA, USA). The data were then analyzed using statistical analysis software (MedCalc version 15.2, MedCalc Software bvba, Ostend, Belgium). Descriptive statistical analysis was performed with means of means to determine the age, ROM, follow-up, and outcome metric scores. Utilizing a random effects model of proportions, we determined the overall rate with its corresponding 95% Confidence Interval (CI) for developing AVN within the overall, the stable, and the unstable cohorts. The same model was utilized to determine the proportion of patients who required a revision procedure who did not develop AVN. These proportions were graphically represented as forest plots (Figure 2).

## 3. Results

A total of 285 patients (287 SCFEs) were managed with a modified Dunn procedure and were included in the study. These children had a mean age of 13.0 years (range of means of 11.9 to 14.3 years) and were followed for a total of 40.4 months (range of means of 12 to 144 months) after the index procedure. Stratifying these hips according to Loder et al., the stable to unstable ratio was skewed toward stable SCFE at 1.5:1 (397 to 258 SCFEs). Of the overall cohort who underwent modified Dunn osteotomies, the femoral heads had a mean slip of 57.3 degrees (range of means of 37 to 72 degrees) as measured by the Southwick angle.

The overall rate of AVN was 14.3% with 95% Confidence Intervals (CI) ranging from 9.3 to 20.2% (Figure 2). Children with stable slips had a lower AVN rate of 10.9% (95% CI: 6.0 to 17.1%, Figure 3), compared with those with unstable slips, in which the rate of AVN increased to 19.9% (95% CI: 12.8% to 28.1%, Figure 4, Appendix A).

After identifying those who did not develop AVN, we determined that 13.3% of these children (95% CI of 8.3% to 19.2%, Figure 5) still required a revision procedure. Of the 51 reported revisions that were not performed as a salvage procedure for AVN, two were due to intra-articular penetration of Kirschner (K) wires, two were the result of clinically significant limb length difference, five were due to postoperative dislocations, and the remaining revisions were indicated due to fixation failure, post-operative dislocations, or correction of impingement (Appendix A).

There was one case of deep surgical site infection that required debridement which ultimately resulted in complete femoral head collapse and ultimately required hip fusion [24]. Of the fixation failures, three patients developed AVN after revision procedures: two cases following revision fixations, and one occurred following a new surgical dislocation and fixation performed for re-displacement. All of the failures of fixation involved screws ranging from 3 to 4.5 mm and/or K wires. Sankar et al. reported that following revision to 6.5 mm cannulated screws four cases of broken implants (4.5 mm screws or heavy-threaded K wires) three revisions were successful, and one developed AVN [22]. Madan et al. used 6.5 mm cannulated screws in all patients, and did not report any implant failures [25].

The overall clinical outcomes were positive. Utilizing the HSS classification for both the stable and unstable cohorts, the mean score of the overall cohort was excellent (>90 points). The overall weighted mean Harris Hip Score at latest follow-up was 93.1 points (range of means of 76 to 99 points). As expected, the stable cohort had a slightly greater weighted mean HHS of 98.9 points (range of means of 85.7 to 98.6 points), and the unstable cohort had a slightly decreased weighted mean HHS of 90.5 points (range of means of 73.2 to 98.3 points).

## 4. Discussion

The optimal surgical treatment for SCFE continues to be controversial. In situ fixation has been a widely used method of treatment for SCFE, but FAI and AVN are concerning complications [3]. The residual deformity due to the lack of anatomical realignment with in situ fixation of SCFE predisposes patients to cam impingement, subsequent FAI, and eventual osteoarthritis of the hip joint [3,6,7,8,10]. Furthermore, studies have reported rates of AVN as high as 47% in cases of unstable SCFE treated with in situ fixation, with most North American reports ranging between 20% to 50% [4,26]. The technique of surgical dislocation and applications of the procedure for conditions such as Legg-Calve-Perthes disease and idiopathic FAI were described by Ganz et al. in 2001 [27]. Ganz et al. supported surgical hip dislocation as a procedure that would allow for full exposure of the femoral head in the surgical treatment of these conditions, without the risk of AVN. The modified Dunn osteotomy, initially described by Leunig et al. maintains perfusion to the epiphysis [13]. More specifically, the soft tissue flap aims to prevent tearing or overextension of retinacular blood vessels during callus debridement of the femoral neck, and thus may prevent epiphyseal perfusion compromise. With this method, epiphyseal perfusion can also be monitored intraoperatively by means of laser Doppler flowmetry [13]. It also allows for anatomic repositioning of the epiphysis on the femoral neck, greater mobility and prevention of FAI. Due to the rarity of the procedure, there has been a paucity of studies evaluating the outcomes and definitively establishing the rate of complications.

By systematically evaluating all current literature, we found that the overall rate of AVN in cases of SCFE treated with a modified Dunn was nearly 14%, with overall excellent clinical outcomes. Comparison of outcome scores across studies is difficult, as selection of outcome scores is not often uniform. Fortunately, a number of studies within this meta-analysis have selected the Harris Hip Score. We identified a weighted mean HHS of 90.5 points (range of means 73.2 to 98.3 points), which compares favorably to HHS outcomes in percutaneous pinning (ranging from 76 to 90).

Reviewing published studies, we noted increased fixation failure with smaller implants with the modified Dunn procedure. It is notable that among the cases of fixation failures described in the studies constituting our meta-analysis, all occurred following fixation with K wires and/or screws with screw caliber less than 6.5 mm. All the fixation failures involved screws ranging from 3 to 4.5 mm and/or K-wires. Sankar et al. reported that following revision for 4 cases of broken implants (4.5 mm screws or heavy-threaded K-wires) with 6.5 mm cannulated screws, 3 revisions were successful, and one developed AVN [22]. Madan et al. used 6.5 mm cannulated screws in all patients, and no implant failures occurred [25].

This study had several limitations. As with all systematic reviews and meta-analyses, the limitations from each specifically assessed study are also our limitations. In addition, the evaluated data were largely heterogeneous, which explains our large confidence intervals. Despite these limitations, this study contains the largest cohort to assess the outcomes of the modified Dunn osteotomy and is important given the recent increase in popularity of this technique for SCFE patients. The authors of the present study believe this study creates a stepping-stone for future randomized controlled studies comparing the modified Dunn osteotomy to in situ pinning for the treatment of SCFEs. In addition, most of the literature describing this topic is published by experts/fellowship trained surgeons, which may oversimplify the complexity of this procedure and may limit the ability for non-experienced surgeons to obtain these positive results.

## 5. Conclusions

We aimed to determine the overall outcomes of the modified Dunn osteotomy as a treatment option for SCFE. After stratifying patients into stable and unstable cohorts, and within these into AVN versus no AVN, and ultimately into those requiring arthroplasty for AVN or revision surgery (despite no AVN); we found overall reduced rates of AVN (as well as overall excellent clinical outcomes as measured by the Harris Hip Score) in patients who underwent this procedure.

This study is critical in the current literature because it suggests that the modified Dunn osteotomy may be a reliable method to treat patients with SCFE. Despite the positive findings in this meta-analysis, it is important to note that this procedure is technically challenging, has a steep learning curve, and is recommended to be performed in hands of experienced surgeons in the proper setting. While further randomized studies are needed before adopting this technique over in situ pinning in selected unstable or severe slips, this study suggests that such future studies are warranted given the acceptable low rates of AVN in patients with unstable SCFE who received this treatment.

## Figures and Tables

**Figure 1 children-09-01680-f001:**
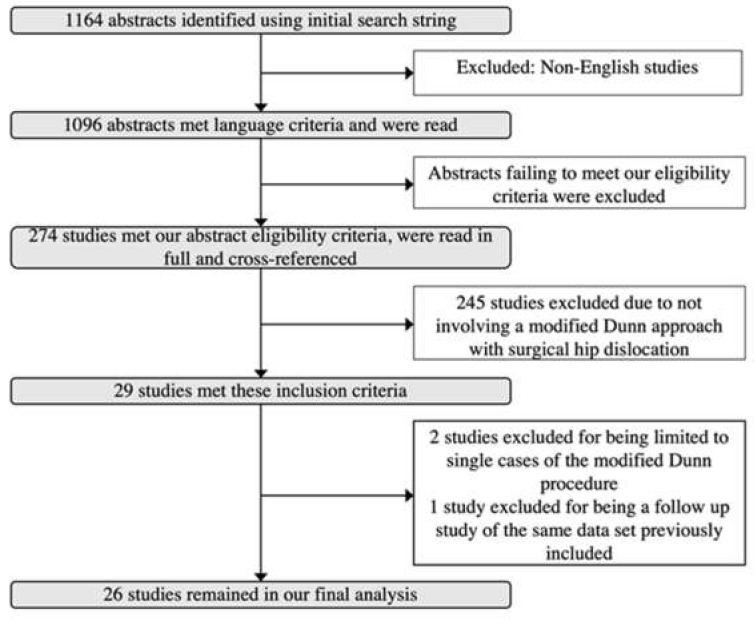
Inclusion Criteria Flowchart.

**Figure 2 children-09-01680-f002:**
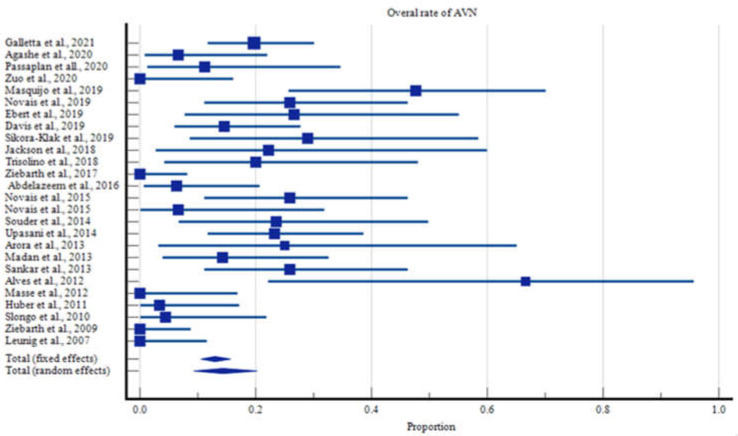
Overall rates of Avascular Necrosis.

**Figure 3 children-09-01680-f003:**
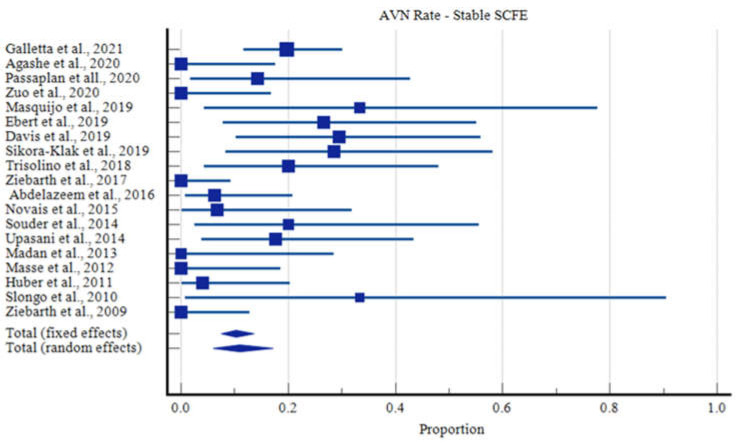
Rates of Avascular Necrosis in Stable SCFE Cases.

**Figure 4 children-09-01680-f004:**
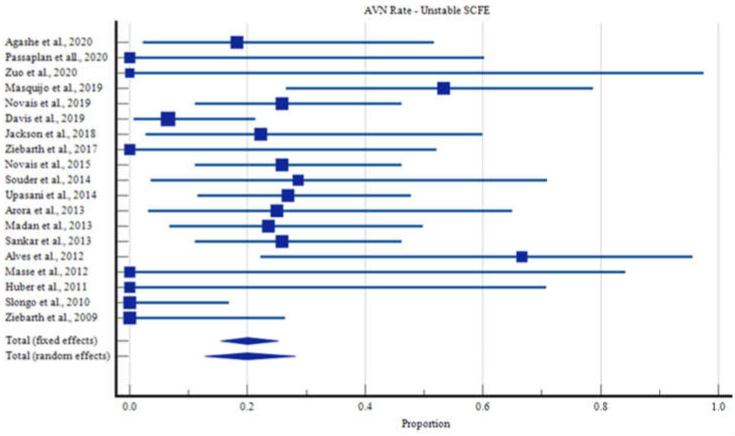
Rates of Avascular Necrosis in Unstable SCFE Cases.

**Figure 5 children-09-01680-f005:**
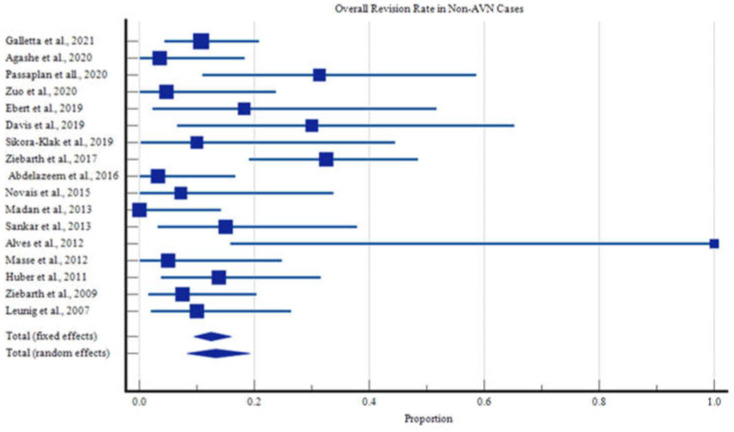
Overall Revision Rates for Reasons Other Than Avascular Necrosis.

## Data Availability

The authors confirm all data, materials, and software support our published claims, and comply with field standards. As this is a meta-analysis of previously published data, all data is available for public scrutiny and thus will not be placed in a unique repository.

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
