# Peer review of "Risk of Avascular Necrosis with The Modified Dunn Procedure in SCFE Patients: A Meta-Analysis"

_children, 2022, doi:10.3390/children9111680_

Round 1

Reviewer 1 Report

It is a fascinating article. It should be read by every orthopedic surgeon who makes this procedure.

On the other hand, I disagree with a sentence in the introduction: " studies have shown associations between percutaneous pinning and an increased risk for developing diabetes, obesity, and hypertension, which may further add to negative outcomes in SCFE treatment" This procedure (percutaneous pinning) couldn`t increase for developing the diabetes/hypertension  - how ??

It well known that obesity or some other diseases could increase the risk of SCFE and then we make the percutaneous pinning.

You have to change it. 

Reviewer 2 Report

This is a well-designed meta analysis on slipped capital femoral epiphysis treatment using a well-known procedure. I want to congratulate the authors for their work, and add some minor suggestions that can increase the strength of their manuscript. Please try to consider the following:

The abbreviation SCFE used on the title your provided might need to be replaced with something that is more reader-friendly for a non-orthopedic journal like Children.

Line 49 - Please define at first what THA means.

A small comment on figures: they seem to be a lower quality and this clearly affects the font on the text. Can the authors provide figures with increased quality and maybe increased font size?
